# Comparison of Immunogenicity and Safety of Inactivated, Adenovirus-Vectored, and Heterologous Adenovirus-Vectored/mRNA Vaccines in Patients with Systemic Lupus Erythematosus and Rheumatoid Arthritis: A Prospective Cohort Study

**DOI:** 10.3390/vaccines10060853

**Published:** 2022-05-26

**Authors:** Theerada Assawasaksakul, Tanat Lertussavavivat, Seelwan Sathitratanacheewin, Nont Oudomying, Preeyaporn Vichaiwattana, Nasamon Wanlapakorn, Yong Poovorawan, Yingyos Avihingsanon, Nawaporn Assawasaksakul, Supranee Buranapraditkun, Wonngarm Kittanamongkolchai

**Affiliations:** 1Division of Rheumatology, Faculty of Medicine, Chulalongkorn University, Bangkok 10330, Thailand; theerada.a@chula.ac.th; 2Division of Nephrology, Faculty of Medicine, Chulalongkorn University, Bangkok 10330, Thailand; 6471016730@student.chula.ac.th (T.L.); yingyos.a@chula.ac.th (Y.A.); 3Department of Medicine, Faculty of Medicine, Chulalongkorn University, Bangkok 10330, Thailand; seelwan@docchula.com (S.S.); nont.oudomying1997@docchula.com (N.O.); supranee.b@chula.ac.th (S.B.); 4Center of Excellence in Clinical Virology, Department of Pediatrics, Faculty of Medicine, Chulalongkorn University, Bangkok 10330, Thailand; preeyaporn.vic@chulahospital.org (P.V.); nasamon.w@chula.ac.th (N.W.); yong.p@chula.ac.th (Y.P.); 5The Royal Society of Thailand, Sanam Sueapa, Dusit, Bangkok 10300, Thailand; 6Cardiometabolic Center, BNH Hospital, Bangkok 10500, Thailand; nawaporn.as@bnh.co.th; 7Center of Excellence in Vaccine Research and Development, Faculty of Medicine, Chulalongkorn University, Bangkok 10330, Thailand; 8Mahachakri Sirindhorn Clinical Research Center, Faculty of Medicine, Chulalongkorn University, Bangkok 10330, Thailand; 9Renal Immunology and Transplantation Research Unit, Faculty of Medicine, Chulalongkorn University, 1873 Rama 4 Rd., Pathumwan, Bangkok 10330, Thailand

**Keywords:** immunogenicity, safety, reactogenicity, SARS-CoV-2 vaccine, SLE, RA

## Abstract

Background: Impaired immune responses to COVID-19 vaccines have been observed in autoimmune rheumatic disease patients. Determining the most effective and safe vaccine regimen is critically needed in such a population. We aim to compare the immunogenicity and safety of three COVID-19 vaccine regimens in patients with systemic lupus erythematosus (SLE) and rheumatoid arthritis (RA). Methods: SLE and RA patients aged 18–65 years who received inactivated (CoronaVac or COVILO), adenovirus-vectored (AZD1222), or heterogeneous (AZD1222/BNT162b2) vaccines were enrolled. Humoral and cellular immune responses were assessed at day 28 after the second vaccination. This was performed using the serum binding antibody level against the receptor-binding domain of the SARS-CoV-2 spike protein (anti-RBD Ig) and IFNy-ELISpot assay (ELISpot), respectively. Reactogenicity was reviewed on day 7 following each vaccination. Disease activity was assessed before and on day 28 after the second vaccination. Results: The cohort consisted of 94 patients (64 SLE and 30 RA). Inactivated, AZD1222, and AZD1222/BNT162b2 vaccines were administered to 23, 43, and 28 patients, respectively. Anti-RBD titers were lowest in the inactivated vaccine group (2.84 AU/mL; 95% CI 0.96–8.44), followed by AZD1222 (233.7 AU/mL; 95% CI 99.0–505.5), and AZD1222/BNT162b2 (688.6 AU/mL; 95% CI 271–1745), *p* < 0.0001. After adjusting for relevant factors, the inactivated vaccine was associated with the lowest humoral response, while adenovirus-vectored/mRNA vaccine was the highest. The proportion of positive ELISpot test was also lowest in the inactivated vaccine group (27%), followed by the adenovirus-vectored vaccine (67%), and the adenovirus-vectored/mRNA vaccine (73%) (*p* = 0.03). All types of vaccine were well-tolerated. There was no flare of autoimmune disease post-vaccination. Conclusion: Adenovirus-vectored and adenovirus-vectored/mRNA vaccines elicited a stronger humoral and cellular immune response than inactivated vaccines, suggesting that they may be more suitable in SLE and RA patients receiving immunosuppressive therapy.

## 1. Introduction

The global COVID-19 pandemic caused by SARS-CoV-2 has infected an estimated 500 million people worldwide, as of April 2022 [1]. Vaccination is one of the most effective strategies to reduce the severity and mortality caused by COVID-19; currently, inactivated, adenovirus-vectored, and mRNA are globally the most widely used vaccine platforms. Evidence has shown that these vaccines are effective and safe in healthy populations [2]; however, impaired vaccine responses have been observed in autoimmune rheumatic disease (AIRD) patients. This has been attributed to the interplay between autoimmune activities along with the patients being on immunosuppressive therapies [3].

Despite strong evidence that AIRD patients elicit a different response to COVID-19 vaccines compared to healthy individuals, data regarding this group of population are limited. Patients with AIRD are often excluded from vaccine studies, leading to limited data available to develop an appropriate vaccination guideline for this vulnerable group of the population. This leads to an important question: “Which type of COVID-19 vaccine is the most suitable for patients with AIRD?” Previous studies have shown that in normal individuals, mRNA vaccine elicits a robust humoral and cellular immune response when compared to the inactivated and adenovirus-vectored vaccines [4], potentially making it an ideal choice for those on immunosuppressive drugs. Nevertheless, there has been no head-to-head comparison of immunogenicity and safety between vaccine types in AIRD patients thus far. The majority of existing studies only examined immune responses to a single vaccine type, either mRNA or inactivated vaccines [5,6,7,8], thus lacking a direct comparison. In addition, despite being one of the most widely used vaccines in the world, data on adenovirus-vectored vaccines in AIRD patients are still lacking.

Vaccine types vary across nations, depending on politics, national health programs, and availability. In Thailand, inactivated and adenovirus-vectored vaccines were the first to become available during the pandemic, followed by the mRNA vaccine. Leveraging the variety of vaccine types used in Thailand, we are able to study the heterogeneity among the immune response towards different types of vaccine types in AIRD patients. In this study, we compared both the humoral and cellular immune responses induced by inactivated, adenovirus-vectored, and heterogeneous adenovirus-vectored/mRNA vaccinated patients with systemic lupus erythematosus (SLE) and rheumatoid arthritis (RA).

## 2. Methods

This is a prospective cohort study investigating the safety and immunogenicity of an inactivated SARS-CoV-2 COVID-19 vaccine (CoronaVac or COVILO), an adenovirus-vectored vaccine (AZD1222, AstraZeneca, Cambridge, UK), and a heterologous regimen of adenovirus-vectored/mRNA vaccine (BNT162b2, Pfizer–BioNTech, New York, NY, USA) in SLE and RA patients. The Institutional Review Board of Chulalongkorn University’s Faculty of Medicine reviewed and approved the trial (882/2021), and it was registered in the Thai Clinical Trial Registry (TCTR20210917003).

### 2.1. Participants

Consecutively, SLE and RA patients aged 18–65 years who met the SLE or RA classification criteria were enrolled. Those with a history of SARS-CoV-2 infection, prior immunization with any SARS-CoV-2 vaccine, allergy to a vaccine component, pregnancy, and active disease at the time of enrollment were excluded. Participants meeting eligibility criteria were invited to participate in the study and provided written informed consent.

### 2.2. Procedure

At baseline, demographic information, current medications, disease activity, and relevant blood samples were collected. The SELENA-SLEDAI and Disease Activity Score 28 ESR (DAS28-ESR) were used to evaluate disease activity for patients with SLE and RA, respectively.

To standardize the various immunosuppressive treatment regimens used in our cohort, the Vasudev score was adopted and modified to calculate the total immunosuppressive load [9]. One unit of immunosuppression was assigned for each of the following doses of immunosuppressive medications: prednisone 5 mg/day, mycophenolate mofetil (MMF) 500 mg/day, azathioprine 100 mg/day, cyclosporine 100 mg/day, tacrolimus 2 mg/day, leflunomide 10 mg/day, and methotrexate 15 mg/week (Table 1). The immunosuppressive unit scale was calculated using the average doses of immunosuppressive medication from baseline to 30 days after the second vaccination.

A vaccine regimen was provided based on patient preference, availability, and hospital policy. For inactivated vaccines, there were two brands available in Thailand, CoronaVac (Sinovac Life Sciences, Beijing, China) and COVILO (Beijing Institute of Biological Products, Beijing, China). Each dose (0.5 mL) of CoronaVac and COVILO contains 600 SU and 3.9–10.4 U of inactivated SARS-CoV-2 virus, respectively. Two doses of vaccine were administered intramuscularly at a 4 week interval. For the adenovirus-vectored vaccine, two doses of AZD1222 (5 × 10^10^ viral particles, 0.5 mL each) were administered intramuscularly with an 8–12 week interval. For heterologous regimen, AZD1222 followed by 30 ug of BNT162b2 were administered intramuscularly with a 4–8 week interval.

### 2.3. Immunogenicity Assessment

On day 28 after completion of the regimen, blood samples were collected to assess immune responses and disease activities. The humoral responses were measured by total serum binding antibody levels against receptor-binding domain (RBD) of the SARS-CoV-2 spike (S) protein (Roche Diagnostics, Basel, Switzerland). Positive values were defined as 0.4 arbitrary U/mL (AU/mL).

The human IFN-γ ELISpot assay (ELISpot) was used to assess cellular immune responses in a random subset of patients in each vaccine group according to the instruction manual (Mabtech, Stockholm, Sweden) [10]. In brief, pre-coated ELISpot plates were washed with phosphate-buffered saline (PBS) and blocked with RPMI1640 medium (Gibco, Waltham, MA, USA) containing 10% heat-inactivated fetal bovine serum (FBS) (Gibco, Waltham, MA, USA) (R10 medium) for 30 min at room temperature. A quantity of 2.5 × 10^5^ PBMCs/well was stimulated for 40 h with SARS-CoV-2 spike peptide pools (S1 and S2) (Mimotopes, Victoria, Australia). Tests were performed in duplicate and with negative control (R10 medium) and positive control (phytohemagglutinin (PHA), Sigma-Aldrich, Burlington, MA, USA). After incubation, the plates were washed with PBS, and a secondary anti-IFN-γ antibody directly conjugated with alkaline phosphatase (7-B6-ALP) was added at the ratio of 1:200 in filtered PBS containing 0.5% FBS for 2 h at room temperature. After washing, the plates were incubated with 100 µL/well of substrate solution (BCIP/NBT-plus) until distinct spots emerged. The reaction was stopped by extensively washing in tap water and rinsing the underside of the membrane, and the plates were left to dry. Inspection and spots counting was performed in the ELISpot reader (ImmunoSpot^®^ Analyzer, Bonn, Germany). Results are expressed as spot-forming cells (SFCs)/10^6^ PBMCs following the subtraction of negative controls. More than 50 spots per 10^6^ peripheral blood mononuclear cells (PBMC) were considered positive.

### 2.4. Safety Assessment

Local and systemic reactogenicity was reviewed on day 7 following each vaccination. Assessment of disease activity was repeated on day 28 after the second vaccination by the previously mentioned scoring system.

### 2.5. Statistical Analyses

Baseline characteristics were reported as mean and standard deviations (SD), or median and interquartile range (IQR), as appropriate. The antibody titer was natural log(ln) transformed as an outcome measure to ensure normality, and was also presented as geometric mean titers (GMT) with a 95% confidence interval (CI). The normalized titer between vaccine groups was analyzed by one-way analysis of variance (one-way ANOVA) and post hoc multiple comparisons by Tukey’s method. Univariable, multivariable linear regression, and multivariable logistic regression analyses were performed to demonstrate the associations between immunogenicity and age, sex, AIRD diagnosis, and dosage of immunosuppressants. Disease activity scores before and after vaccination were compared using paired T-test. The percentage of cellular responsiveness and reactogenicity were compared among vaccine types using the chi-square statistic. All statistical analyses had α levels of <0.05 for defining significance. Data were statistically analyzed using JMP software V.13.2.1 (SAS Institute, Cary, NC, USA) and graphs were created using GraphPad Prism V.4.03 (GraphPad Software, La Jolla, CA, USA).

## 3. Results

The cohort consisted of 94 patients, with 64 (68%) having SLE and 30 (32%) having RA. Ninety-three percent of the patients were female. The inactivated, AZD1222, and AZD1222/BNT162b2 vaccines were administered to 23 (8 CoronaVac and 15 COVILO), 43, and 28 patients, respectively. The baseline characteristics, such as gender, disease duration, underlying autoimmune disease, and disease activity, were comparable between the vaccine groups. AZD1222 was primarily administered to the elderly in Thailand, reflected by the average age of those who received it being the highest among the vaccine types (Table 2). In addition, the immunosuppressive load among those who received AZD1222 was less than those with other vaccine types.

### 3.1. Humoral Immune Responses

The inactivated vaccine group had the lowest seroconversion rate (52%) compared to AZD1222 (93%) and AZD1222/BNT162b2 (96%), *p* < 0.0001. The anti-RBD titers were also lowest in the inactivated vaccine group (GMT 2.84 AU/mL; 95% CI 0.96 to 8.44), followed by AZD1222 (GMT 233.7 AU/mL; 95% CI 99.0 to 505.5), and AZD1222/BNT162b2 (GMT 688.6 AU/mL; 95% CI 271 to 1745), *p* < 0.0001 (Figure 1A). After adjusting for age, gender, diagnosis, and immunosuppressive load, the vaccine regimen remained an independent predictor of anti-RBD titer (Table 3). The inactivated vaccine was associated with the lowest humoral response, while AZD1222/BNT162b2 was the highest. The immunosuppressive load was also negatively associated with anti-RBD titer (beta = −0.38; 95%CI −0.66 to −1.0, *p* = 0.008) (Figure 2).

### 3.2. Cellular Responses

The cellular immune response was assessed in 41 patients. In the inactivated vaccine, AZD1222, and AZD1222/BNT162b2 groups, there were 15, 15, and 11 patients, respectively. The baseline characteristics, including age, gender, underlying rheumatologic disease, and immunosuppressive load, were comparable between the vaccine groups (*p* > 0.05). The proportion of patients who had a positive ELISpot test was lowest in the inactivated vaccine group, followed by the AZD1222 and the AZD1222/BNT162b2 vaccine groups (27%, 67%, and 73%, respectively, *p* = 0.03). The mean ELISpot levels of each vaccine group also followed a similar trend (inactivated vaccine 39.73 (95% CI 11.41 to 68.06), AZD1222 176.7 (95% CI 70.98 to 282.4), and AZD1222/BNT162b2 907.1 (95% CI 358.9 to 1445), *p* < 0.0001) (Figure 1B). ELISpot level was weakly correlated with anti-RBD titer (r-square = 0.37, *p* < 0.0001, Appendix A).

### 3.3. Safety, Reactogenicity, and Disease Activity

Most patients (90%) experienced at least one non-serious adverse reaction in both injections. Overall, the most reported adverse reaction was injection site pain (36%), followed by fatigue (21%) and fever (21%). Patients who had AZD1222 and AZD1222/BNT162b2 reported more injection site pain than for the inactivated vaccine (41.3%, 58.6% and 10.8%, respectively; *p* < 0.001).

A total of 34 SLE (48%) and 30 RA patients (90%) had complete data on disease activity scores pre- and post-vaccination. There was no difference in the change of SLEDAI score across the vaccine regimens in patients with SLE (95%CI −0.94 to 0.94; *p* > 0.999). DAS28-ESR was lower in patients with RA after vaccination (95%CI −1.12 to −0.17; *p* = 0.01) (Appendix A).

During the study period, two patients were infected with COVID-19. One patient was given AZD1222/BNT162b2 and developed COVID-19 infection one day after the second dose. The other received AZD1222 and had COVID-19 infection 29 days after the first dose.

## 4. Discussion

Despite the global strategy to use vaccination to limit the spread of COVID-19, data on their efficacy and adverse effects in individuals with AIRD are under-reported, because the individuals were often omitted from clinical studies that led to the vaccines’ approval [11,12,13,14]. In AIRD patients, an effective COVID-19 vaccine strategy is critical because altered vaccine responsiveness makes this population vulnerable to COVID-19 infection and severe complications [15].

The anti-RBD titer is a valuable measure for estimating individual resistance to COVID-19 infection [16]. A high anti-RBD titer also provides cross-protection against SARS-CoV2 variants of concern [16]. In comparison, among the three vaccine regimens, we found that only half of the AIRD patients who received inactivated vaccines had seroconversion, whereas almost all of the patients who received AZD1222 and AZD1222/BNT162b2 were seroconverted. The mean anti-RBD titer of inactivated vaccine recipients was also lower than AZD1222 and AZD1222/BNT162b2 by 82 and 242 fold, respectively. Our findings were consistent with studies performed on the healthy population, where lower humoral response was reported in inactivated vaccine when compared to adenovirus-vectored or mRNA vaccine [17,18]. The other studies also found that heterologous adenovirus-vectored/mRNA vaccination was more immunogenic than homologous adenovirus-vectored vaccination [19,20]. Vaccine type remains to be a strong independent predictor of humoral immune response in AIRD patients, even when it is adjusted for other potential confounders, including baseline characteristics and immunosuppressive load.

Cellular immunity plays a pivotal role in long-term protection as well as reducing the severity of COVID-19 infection, while antibody titers wane over time [21]. It also cross-reacts with the variants of concern that can evade neutralizing antibodies [22,23]. Nevertheless, there were only a few studies regarding cellular immunogenicity to mRNA vaccines in AIRD patients [6] and none for the inactivated and adenovirus-vectored vaccines. Our study demonstrated that there was a weak association between anti-RBD titer and ELISpot level, and the pattern of cellular immune response of each vaccine type followed that of the humoral immune response. The proportion of patients with positive cellular immunity was lowest in inactivated vaccine recipients when compared to AZD1222 and AZD1222/BNT162b2.

Apart from the above findings, we also demonstrated an independent association between high immunosuppressive load and poor humoral immune response. A high potency vaccine should be strongly considered in AIRD patients to provide adequate humoral and cellular protection, especially for those who received a high immunosuppressive load.

Overall, every type of vaccine is well tolerated with only mild adverse reactions such as injection site pain or fever. The inactivated vaccine had the least local and systemic adverse reaction compared to the others. A finding from a cross-sectional study in Mexico evaluating adverse events of six COVID-19 vaccines in 225 AIRD patients was consistent with our study [24]. There was no evidence of disease flares upon using any type of vaccine in our cohort, and no change in disease activity score was observed. Similar to our findings, other studies reported low incidence of AIRD flares in those receiving inactivated, adenovirus-vectored, and mRNA vaccines [8,25,26,27].

Limitations regarding the subject of our study include the number of participants, focusing primarily on SLE and RA patients and excluding other immunosuppressive regimens. Consequently, our results may not apply to other autoimmune diseases or those who received different types of immunosuppressive regimens, such as immunomodulatory biologic agents. Due to the limited sample size, we combined two brands of inactivated vaccines in the analysis, although their efficacy may not be equivalent. Our results may also not be applicable to those receiving homologous mRNA vaccines. It is also important to note that the immunosuppressive load calculation used within this research was adapted from a previous study that was based on arbitrary criteria. Furthermore, using anti-RBD seroconversion and ELISpot for T cell response might not fully reflect protection against the disease. However, there is mounting evidence that the levels of antibodies directed against the RBD domain of the spike protein measured in our study are a good predictor of vaccine efficacy [28].

## 5. Conclusions

Different regimens of the COVID-19 vaccine are not equal in terms of immunogenicity. Most of them are safe and well-tolerated with no flare of autoimmune disease post-vaccination. Our study highlighted the urgent need for a booster dose of a high potency vaccine in patients with AIRD, particularly those who have previously received an inactivated vaccine or received high doses of immunosuppressants.

## Figures and Tables

**Figure 1 vaccines-10-00853-f001:**
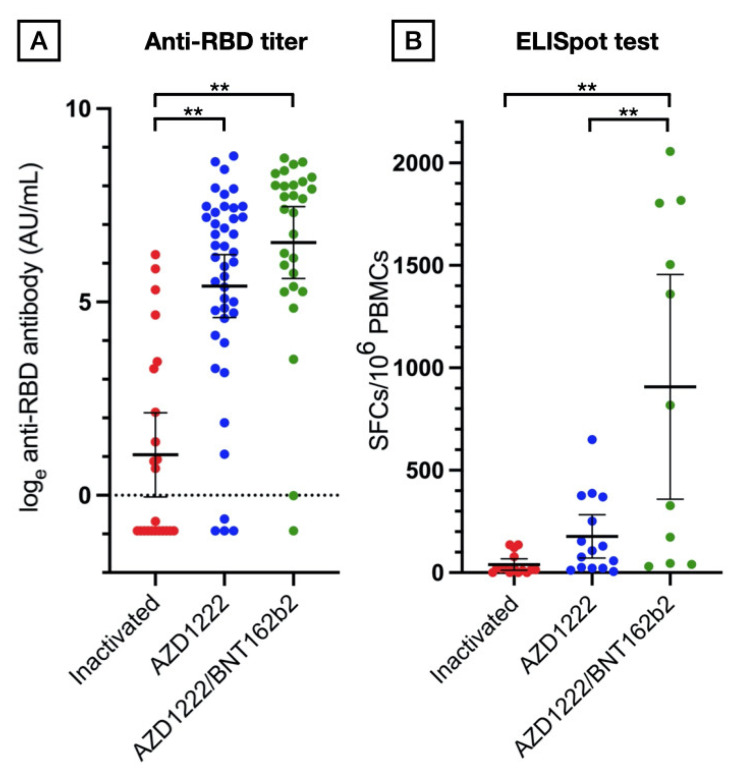
Immunogenicity assessment stratified by vaccine group. (**A**) Scatter plot of natural log-transformed total immunoglobulin specific to the receptor-binding domain (RBD) in inactivated, AZD1222, and AZD1222/BNT162b2 vaccine groups after two doses. (**B**) Scatter plot of spot-forming cells (SFCs)/10^6^ peripheral blood mononuclear cells (PBMCs) after a two dose completion stratified by vaccine group. Data points are the reciprocals of the individual. Line indicates mean and bar indicates 95% confidence interval. ** indicates *p* < 0.001.

**Figure 2 vaccines-10-00853-f002:**
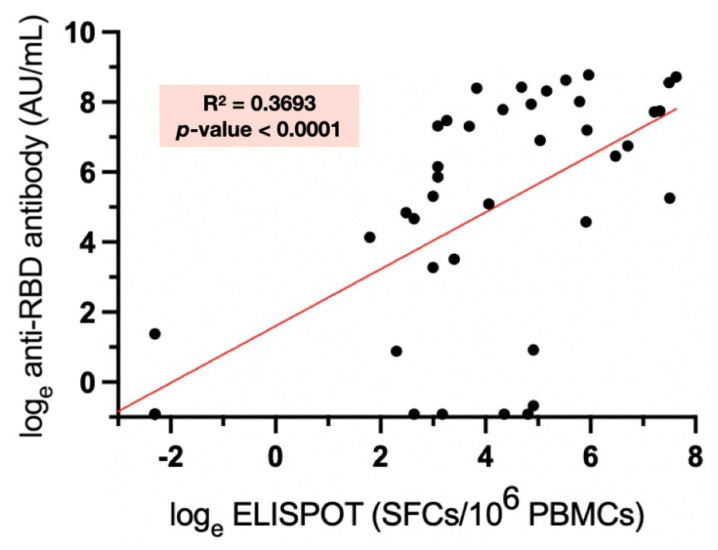
Relationship between total immunosuppressive load and natural log-transformed total anti-RBD Ig level at 28 days after the second vaccination.

**Table 1 vaccines-10-00853-t001:** A scale modified from Vasudev et al. to quantify the immunosuppressive medication (9). One unit of immunosuppression was assigned to the corresponding doses of agents.

Immunosuppression Units	Agents	Unit Dose
1	Prednisone	5 mg/day
1	Azathioprine	100 mg/day
1	Cyclosporine	100 mg/day
1	Tacrolimus	2 mg/day
1	Mycophenolate mofetil	500 mg/day
1	Methotrexate	15 mg/week
1	Leflunomide	10 mg/day

**Table 2 vaccines-10-00853-t002:** Baseline characteristics stratified by vaccine groups.

Characteristic	AZD1222 (*n* = 43)	AZD1222/BNT162b2 (*n* = 28)	Inactivated (*n* = 23)	*p*-Value
Age (year)	49.5 (12.5)	44.8 (10.1)	38.9 (13.2)	0.004
Female gender (%)	40 (93%)	26 (93%)	21 (91%)	1
Duration of disease (year)	10.2 (9.7)	11.9 (7.8)	8.3 (8.8)	0.4
Underlying autoimmune disease (%)				
- RA	15 (35%)	9 (32%)	6 (26%)	
- SLE	28 (65%)	17 (68%)	17 (74%)	0.8
Disease activity				
- SLEDAI	3.14 (2.95)	2.11 (2.35)	3.12 (2.74)	0.4
- DAS28	3.59 (1.68)	3.84 (1.52)	2.47 (0.72)	0.2
- CDAI	12.1 (13.6)	17.2 (13.1)	3.3 (5.4)	0.1
Immunosuppressant load (unit)	2.42 (1.61)	2.66 (1.68)	3.62 (2.33)	0.04
Immunosuppressive therapy (%)				
Mycophenolate mofetil	21 (45.7%)	14 (48.3%)	16 (43.2%)	
Azathioprine	4 (8.7%)	3 (10.3%)	4 (10.8%)	
Tacrolimus	0 (0%)	0 (0%)	3 (8.1%)	
Cyclosporine	4 (8.7%)	1 (3.4%)	2 (5.4%)	
Methotrexate	14 (30.4%)	8 (27.6%)	12 (32.4%)	
Leflunomide	3 (6.5%)	3 (10.3%)	4 (10.8%)	
Prednisolone	36 (78.3%)	23 (79.3%)	32 (86.5%)	
Antimalarial	30 (65.2%)	19 (65.5%)	25 (67.6%)	

Data presented as mean (SD) unless otherwise noted. The immunosuppressant load was scaled using Vasudev score (see method). RA: Rheumatoid arthritis; SLE: Systemic lupus erythematosus.

**Table 3 vaccines-10-00853-t003:** Linear regression models for natural log-transformed anti-RBD antibody at 28 days after the second vaccination.

	Univariable			Multivariable		
Variable	Regression Coefficient	95% CI	*p*-Value	Regression Coefficient	95% CI	*p*-Value
Age	0.02	0.04 to 0.14	0.0006	0.02	−0.03 to 0.07	0.4
Female	0.33	−0.96 to 1.62	0.6	0.38	−0.56 to 1.32	0.4
Diagnosis						
RA	Ref	Ref		Ref	Ref	
SLE	−0.82	−1.53 to −0.11	0.02	−0.59	−1.22 to 0.03	0.06
Vaccine group						
AZD1222	Ref	Ref		Ref	Ref	
Inactivated	−3.28	−4.10 to −2.47	<0.0001	−2.82	−3.63 to −2.01	<0.0001
AZD1222/BNT162b2	2.2	1.43 to 2.98	<0.0001	2.09	1.36 to 2.81	<0.0001
Immunosuppressant load	−0.64	−0.98 to −0.30	0.0003	−0.38	−0.66 to −1.0	0.008

The immunosuppressant load was scaled using Vasudev score (see method). CI: Confidence interval; RA: Rheumatoid arthritis; SLE: Systemic lupus erythematosus; Ref: Reference category in regression model.

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
