# Peer review of "Comparison of Immunogenicity and Safety of Inactivated, Adenovirus-Vectored, and Heterologous Adenovirus-Vectored/mRNA Vaccines in Patients with Systemic Lupus Erythematosus and Rheumatoid Arthritis: A Prospective Cohort Study"

_vaccines, 2022, doi:10.3390/vaccines10060853_

Round 1

Reviewer 1 Report

In the present study authors have compared the humoral and cellular immune responses induced by inactivated (CoronaVac or COVILO), adenovirus-vectored (AZD1222, AstraZeneca) and heterogeneous adenovirus-vectored /mRNA (AZD1222/BNT162b2, Pfizef-BioNTech) vaccines in Thai adult patients with systemic lupus erythematosus (SLE) or rheumatoid arthritis (RA). Result show that inactivated vaccine was associated with the lowest humoral response, while adenovirus vectored/mRNA vaccine was the highest. Immune cellular response was also lowest in the inactivated vaccine group, followed by the adenovirus-vectored vaccine and then the adenovirus-vectored/mRNA vaccine. Moreover, the three types of vaccines were well tolerated, with no flare of the disease or increase in disease activity scores post-vaccination.

The manuscript is a well performed scientific research, with clear objectives, well stated methodology and therefore reliable results that correlate and confirm previous studies. The discussion is brief, clear and supported by the results.

However, some questions have arisen while reading the manuscript:

  1. In lines 52-53 author say “SARS-CoV-2 has infected an estimated 500 million people worldwide”. Although this is true when writing the manuscript, when it is published or read the numbers may have changed greatly. Therefore, it should state the date of the data (“…as of April 2022”, for example).
  2. Authors do not compare an mRNA only vaccine together with the other three types, despite they say that these were the second kind of vaccines available in Thailand during the pandemic. Why is that? The results would have been clearer if a pure mRNA vaccine had been tested, instead of a heterogeneous one. Authors do not explain this omission nor do the election of the heterogeneous vaccine and this deserves to be clarified.
  3. The dose of the two inactivated vaccines seem quite different (600SU and 3.9-10.4 U of inactivated SARS-CoV-2 virus for CoronaVac and COVILO respectively). As COVILO was administered to 15 patients and the other one to just 8, can the different composition of the inactivated vaccines have affected the final results? In other words, is it possible that if one of the vaccines was less active, because it had less virus particles, it can have lowered the results for these vaccines?
  4. In line 167, when talking about the inactivated vaccines, Sinovac is mentioned instead of CoronaVa. Is it similar, the same vaccine or is it a mistake? It should be clarified.
  5. Why is that “immunosuppressive load among those who received AZD1222 was less than those with other vaccine types”? (lines 172-173). Is it due to a treatment protocol or was it observed by chance? In the latter case, which could be the reason? Moreover, this can have increased the results for this kind of vaccine.
  6. Related to the above point, knowing the immunosupressant load mean for each group or vaccine might have brought interesting results, because, as shown in Figure 2, it clearly affects the anti-RBD titer. Was it not possible to perform this calculation?
  7. In the Discussion section, it is said “Heterologous adenovirus-vectored /mRNA vaccination was also more immunogenic than homologous adenovirus-vectored vaccination and comparable to homologous mRNA vaccination (19, 20)”. Although it refers to different articles, maybe this sentence could be rephrased because now it seems that in fact a homologous mRNA vaccine was tested in this study, and it is not the case.

Apart from these small commentaries and questions, the manuscript seems coherent and well founded to achieve the conclusions stated.

Reviewer 2 Report

This prospective heat-to-head study is overall well designed and conducted and it takes into account the immunosuppressive load according to the treatment.

Interestingly, the authors inform about a weak association between anti-RBD titers and ELISpot positiveness. Could be that analysis shown?. There is evidence that T cells may be the major mediators in the related infections MERS and SARS-CoV-1, even high antibody levels have been associated with increased inflammation and impaired clinical outcome in the last case. Thus, it may be interesting to know also to what extent each immunosuppressive drug affects cellular immunity and if changes in certain antirheumatic medication regimens might increase the effectiveness.

Table 2: How the % of immunosuppressive therapy were calculated? The sum in each column is above 100. Were some patients treated with several immunosuppressant drugs? Please, clear this point.

Some questions regarding the cellular response study need to be cleared: what were used as negative controls? (no peptides; blood before vaccination; other…); Why is the chi-squared statistic used to compare the percentage of cellular responsiveness? How was calculated that percentage? (in figure 1B the data are indicated as SFCs/106 PBMCs).

For the association analysis, why the strength of the association of each variable with the response was not estimated with the odds ratio (OR)?

As the study did not include the immune response after the first dose, it is not possible to know how develops the primary immune response during the primming and the recall response separately and compare the AZD1222 and AZD1222/BNT162b2 to find out potential benefits from the heterogeneous regimen.

Another point is the immune state of the enrolled AIRD patients. The overall load of immunosuppression is determined not only by immunosuppressive drugs but also by the humoral and cellular immunity of the recipient. For a more accurate definition of immunosuppression, a flow cytometry study of immune B and T cell profiles (naive, memory, exhausted) could be conducted.

Regarding safety, all of the three vaccine regimens seemed to be safe. Indeed, a register with a great number of patients from autoimmune rheumatic and musculoskeletal diseases indicates that the COVID-19 vaccines Pfizer/BioNTech vaccine (70%), AstraZeneca/Oxford (17%) and  Moderna (8%) were well-tolerated with rare reports of I-RMD flare and very rare reports of serious Aes (Safety of vaccination against SARS-CoV-2 in people with rheumatic and musculoskeletal diseases: results from the EULAR Coronavirus Vaccine (COVAX) physician-reported registry, http://dx.doi.org/10.1136/annrheumdis-2021-221490).

Minor issues: some text lines in tables 2 and 3 must be aligned; in Table 3: the meaning for “Ref” needs to be indicated.
